# Wildfires disproportionately affected jaguars in the Pantanal

Alan Eduardo de Barros [1✉], Ronaldo Gonçalves Morato [2], Christen H. Fleming[3,4], Renata Pardini[5], Luiz Gustavo R. Oliveira-Santos[6], Walfrido M. Tomas[7], Daniel L. Z. Kantek[8], Fernando R. Tortato[9], Carlos Eduardo Fragoso[10], Fernando C. C. Azevedo[11,12], Jeffrey J. Thompson [13,14] & Paulo Inácio Prado[1]

The Pantanal wetland harbours the second largest population of jaguars in the world. Alongside climate and land-use changes, the recent mega-fires in the Pantanal may pose a threat to the jaguars' long-term survival. To put these growing threats into perspective, we addressed the reach and intensity of fires that have affected jaguar conservation in the Pantanal ecoregion over the last 16 years. The 2020 fires were the most severe in the annual series, burned 31% of the Pantanal and affected 45% of the estimated jaguar population (87% of these in Brazil); 79% of the home range areas, and 54% of the protected areas within home ranges. Fires consumed core habitats and injured several jaguars, the Pantanal's apex predator. Displacement, hunger, dehydration, territorial defence, and lower fecundity are among the impacts that may affect the abundance of the species. These impacts are likely to affect other less mobile species and, therefore, the ecological stability of the region. A solution to prevent the recurrence of mega-fires lies in combating the anthropogenic causes that intensify drought conditions, such as implementing actions to protect springs, increasing the number and area of protected areas, regulating fire use, and allocating fire brigades before dry seasons.

[1] Instituto de Biociências, Departamento de Ecologia, Universidade de São Paulo, Rua do Matão, Trav. 14, no. 321, Cidade Universitária, São Paulo, SP 05508-090, Brazil. [2] Centro Nacional de Pesquisa e Conservação de Mamíferos Carnívoros, Instituto Chico Mendes de Conservação da Biodiversidade, Atibaia, SP 12952011, Brazil. [3] Department of Biology, University of Maryland College Park, College Park 20742 MD, USA. [4] Smithsonian Conservation Biology Institute, National Zoological Park, 1500 Remount Road, Front Royal 22630 VA, USA. [5] Instituto de Biociências, Departamento de Zoologia, Universidade de São Paulo, Rua do Matão, Trav. 14, no. 321, Cidade Universitária, São Paulo, SP 05508-090, Brazil. [6] Department of Ecology, Federal University of Mato Grosso do Sul, Campo Grande, MS 79070-900, Brazil. [7] Empresa Brasileira de Pesquisa Agropecuária (Embrapa Pantanal), Corumbá, MS, Brazil. [8] Instituto Chico Mendes de Conservação da Biodiversidade (ICMBIO), Estação Ecológica de Taiamã (EET), Cáceres, MT, Brazil. [9] Panthera, 8 West 40th Street, 18th Floor, New York, NY, USA. [10] Associação Onçafari, Rua Ferreira de Araújo, 153, Conjunto 14, Sala 4, Pinheiros, 05428-000 São Paulo, SP, Brazil. [11] Departamento de Ciências Naturais - Universidade Federal de São João del Rei., São João Del Rei, MG, Brazil. [12] Instituto Pró-Carnívoros Atibaia, Av. Horácio Neto, 1030, 12954–010 Atibaia, SP, Brazil. [13] Instituto Saite, Asunción, Paraguay. [14] Asociación Guyra Paraguay and CONACYT, Parque Ecológico Asunción Verde, Asunción, Paraguay. ✉email: alanbiology@gmail.com

The jaguar (*Panthera onca*) has been considered as Near Threatened for a quarter century[1]. Although several subpopulations have already been recognized as endangered or critically endangered[1–4], some stability is still assumed within the Amazon and Pantanal biomes[1–3,5]. The Pantanal is a biodiversity/ecosystem services hotspot[6,7] and was declared a National Heritage Site by the Brazilian Constitution of 1988 and a Biosphere Reserve by UNESCO in 2000[7,8]. However, jaguar population is increasingly threatened in the Pantanal[1,5] due to the accelerated intensification of land use within the biome and adjacent areas. The main threats to jaguar conservation are habitat loss[4], prey poaching[4], retaliation for livestock depredation[9–11], pollution from mining and pesticides[12], increased agricultural activities[13] and human infrastructure (e.g., increased number of dams and roads[14–16]). Although fire is typically considered a threat to a small proportion of the overall jaguar population[1], the unprecedented severity of the 2020 fires in the Pantanal[17–21] suggests that fire may be an unaccounted risk to jaguar conservation in this biome.

An unusual number of fires started in the 2019 wet season in the Pantanal, which intensified in the following dry season[17–22] (Figs. S1, S2). In the Brazilian Pantanal, these fires reached 40,000 km², with a recorded number of fire outbreaks 400% greater than the median between 1998–2019[20]. Human-related ignitions[17,18,21] combined with a large amount of flammable biomass resulting from a severe drought[22–25] (Fig. S3–S9) fuelled extensive fires that spread underneath the soil and crossed through areas that are usually flooded or close to water[17,23].

The fires consumed considerable portions of forest cover and ecologically important areas that would otherwise provide shelter, food, and landscape connectivity to many species[17–19], directly killing about 17 million vertebrates[26]. Furthermore, the fires impacted biological communities in the Pantanal beyond the affected land extent. For example, the fires destroyed extensive swathes of private and public protected areas (PAs)[17,27] (Fig. S20), forest patches at high elevation areas, riparian vegetation, and keystone tree species that provide fruits or nesting sites for birds (e.g., for Hyacinth macaw, *Anodorhynchus hyacinthinus*)[17,23]. Plants with low resistance and resilience against fire[28,29] and less agile vertebrates such as anteaters, armadillos, sloths, and reptiles[30] were probably the most affected species.

Despite jaguars' speed and ability to move large distances, several individuals were injured during the 2020's mega-fire. Some rescued animals were unable to return to the wild because of the gravity of their injuries[19], and at least two rescued individuals died. Moreover, studies in the Pantanal showed short[31] and long-term negative effects of fires on gross primary productivity (GPP)[32]. Although the opposite trend has been pointed in the long term for Amazon[33] and temperate forests[34]. Nonetheless, in high GPP areas, such as the Pantanal, jaguars have smaller home ranges[35] and thus occur at higher densities[36]. Given recent and projected increases in global and regional temperatures[37,38], the recurrence of extreme droughts[8,22] and uncontrollable fires[17–19,23] may reduce overall productivity and impact jaguar movements patterns, space use, and habitat selection.

While the consequences of 2020 Pantanal fires warrant further studies, determining the disproportionate impact of these human-induced fires on critical species is the first step in understanding the extent and severity of the damage. Here, we addressed this challenge by investigating how fire has impacted estimated jaguar numbers, areas selected as home ranges (hereafter HRs), and priority areas for conservation of the jaguar, an umbrella species and apex predator in the Pantanal ecoregion (Brazil, Bolivia, and Paraguay). Home ranges are areas in which an animal expend their lives[39], which can be measured based on location and temporal distribution[40], and also be assumed as jaguar priority areas (2nd order habitat selection)[41]. The rationale here, is that if conditions were similar, an area selected as a HRs by a resident jaguar will more likely be occupied by another resident jaguar than a random area.

In order to assess the annual impact of fires (2005–2020)[42,43] on jaguars, we used two main sources of data, as follows: a) published estimates of jaguar abundance for its entire geographic range based on spatial predictions of density and distribution[36] and b) home range (HRs) areas estimated for 48 resident jaguars monitored between 2005 and 2016 in the Pantanal[44] which we assumed as stable jaguar priority areas (2nd order habitat selection)[41] in order to compare them over the entire evaluated period (2005–2020). We adopted an approach similar to that of a study investigating the impacts of deforestation and fires on jaguars in the Amazon[45]. We used jaguar density estimates[36] in areas overlapping with the occurrence of fire gauges as a proxy for the number of animals potentially displaced, injured, or killed by fires[45]. Home-range areas were estimated from GPS tracking data[44] of 45 jaguar individuals tracked in the Brazilian Pantanal and three in the Paraguayan and Bolivian Pantanal between 2005 and 2016. We selected only individuals whose HRs could be assumed as stable areas, capable of maintaining a resident animal[39–41] or likely to be occupied by a new individual if conditions were kept similar.

We compared remote sensing data on fires that occurred in the last 16 years in the Pantanal to investigate temporal trends of fire affecting (I) the number of jaguars, (II) the proportion and extent of areas selected as home ranges (HRs) by jaguars, and (III) the proportion and extent of legally protected areas (PAs) within the HR of individual jaguars. We focused primarily on assessing the impact of fires on jaguars rather than investigating their causes. However, we included extensive previous evidence to discuss potential causes, impact mitigation, and biodiversity conservation in the Pantanal biome (SI).

## Results

Fire occurrences increased with drought conditions from 2019 to 2020 (Figs. 1, 2). Notably, the 2020 Pantanal fires exhibited the highest mean intensity of the period (352.3 K), 16 Kelvin higher than the median of the previous 15 years (Fig. 3). Fires affected 31% of the Pantanal, most of it in Brazil (87% of the total burnt area), corresponding to 33% of the legal boundaries of the Brazilian biome. These estimates were highly correlated with estimates of the impact of fires in the Pantanal based on other fire datasets[20,46] (R > 0.93, see Figs. S1, S2). However, the severity of the damage reached higher proportions in critical areas for jaguar conservation, drastically affecting their estimated numbers and HRs and burning 62% of the PAs in the Brazilian Pantanal (Figs. 1–3).

**Impacts of fires on jaguars in 2020**. Based on the spatial congruence of raster layers exhibiting fire occurrence (or intensity), jaguar population densities, HRs, and PAs within HRs, we found that:

(I) The impact of fire on jaguar population was exceptionally high in 2020 because the fires coincided spatially with areas of high population density[36] (Fig. 1, Table 1). Fires reached 45% (n = 746 individuals) of the estimated jaguar population throughout the Pantanal (n = 1668 individuals). This figure is 3.3 times the median of 15 previous years, if we use the same data on population densities and yearly data on area affected by fires. The Brazilian Pantanal had the highest proportion of jaguars

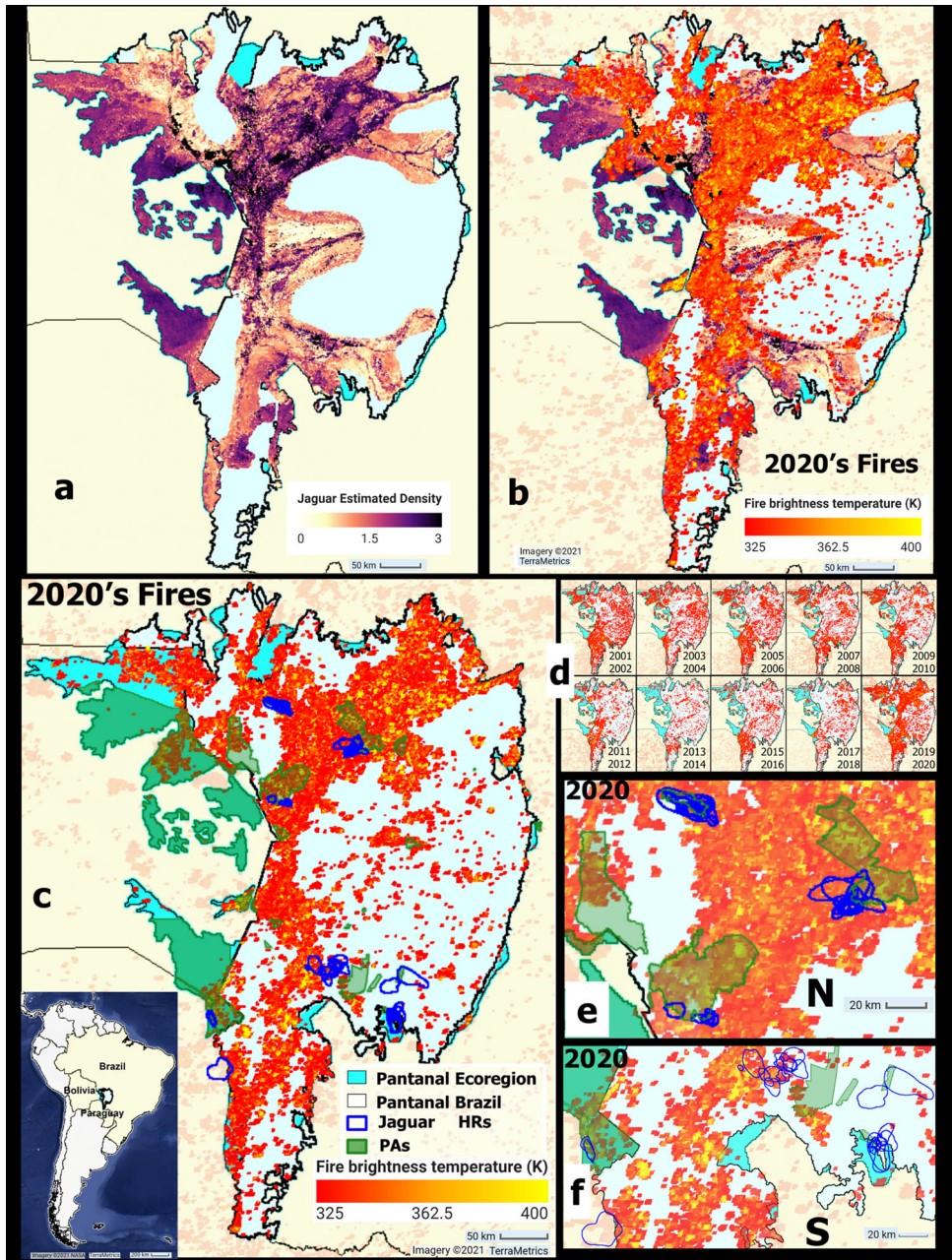

**Fig. 1 Maps showing the location of the Pantanal and the impact of the 2020 fires on jaguars. a** Adjusted jaguar density estimates[36]/100 km$^2$ used as a proxy for the number of jaguars in the Pantanal. **b** Impact of the 2020 fires[42,43] on jaguar estimates. **c** Impact of 2020 fires on jaguar[44] home ranges (HRs) and Protected areas (PAs)[92-97]. **d** Biennual impacts of fires since 2001. Northern (**e**) and southern Pantanal (**f**) zoomed-in detail. The PAs are represented in green and HRs of resident jaguars in blue. Fire occurrence and its corresponding fire brightness temperature are represented in the scale bar from red to yellow (brightest).

affected by fires (87%, $n = 649$ individuals), followed by the Bolivian (12%) and the Paraguayan Pantanal (1%).

(II) The year 2020 exhibited the highest proportion and extent of jaguar HRs burnt by fires in 16 years, when 38 out of the 48 HRs (79%) were affected (Figs. 2, 3). The median burnt extent in jaguar HRs was 78%, corresponding to 2,718 km$^2$. We also documented the highest mean fire intensity, five times higher than the estimated median for the previous 15 years. Significant impacts occurred in the northern Pantanal (Figs. 1, 3), where 2,098 km$^2$ of HRs were affected, corresponding to a median extent of 97% of the HR area (mean = 87%). We note that the inferences about the fires in the HRs throughout the Pantanal

were based mainly on jaguars tracked in Brazil ($n = 45$) and only three in Paraguay/Bolivia.

(III) The 2020 fires affected 78% of the total area of the jaguar HRs overlapping with protected areas. Home ranges covered 1,354 km$^2$ of PAs where fires burned 1,054 km$^2$, an area 9.2 times greater than the median area (of PAs within HRs) burned during the previous 15 years in the entire Pantanal. In Brazil, the area of PAs burned within HRs totalled 970 km$^2$ (72%). Fires occurred in 54% ($n = 26$) of the HRs with PAs, with a median extent of burned PAs of 94%. The impacts of fire on PAs were particularly high in the northern Brazilian Pantanal, where the mean PA burned within HRs reached 91% (median=100%) (Fig. 3).

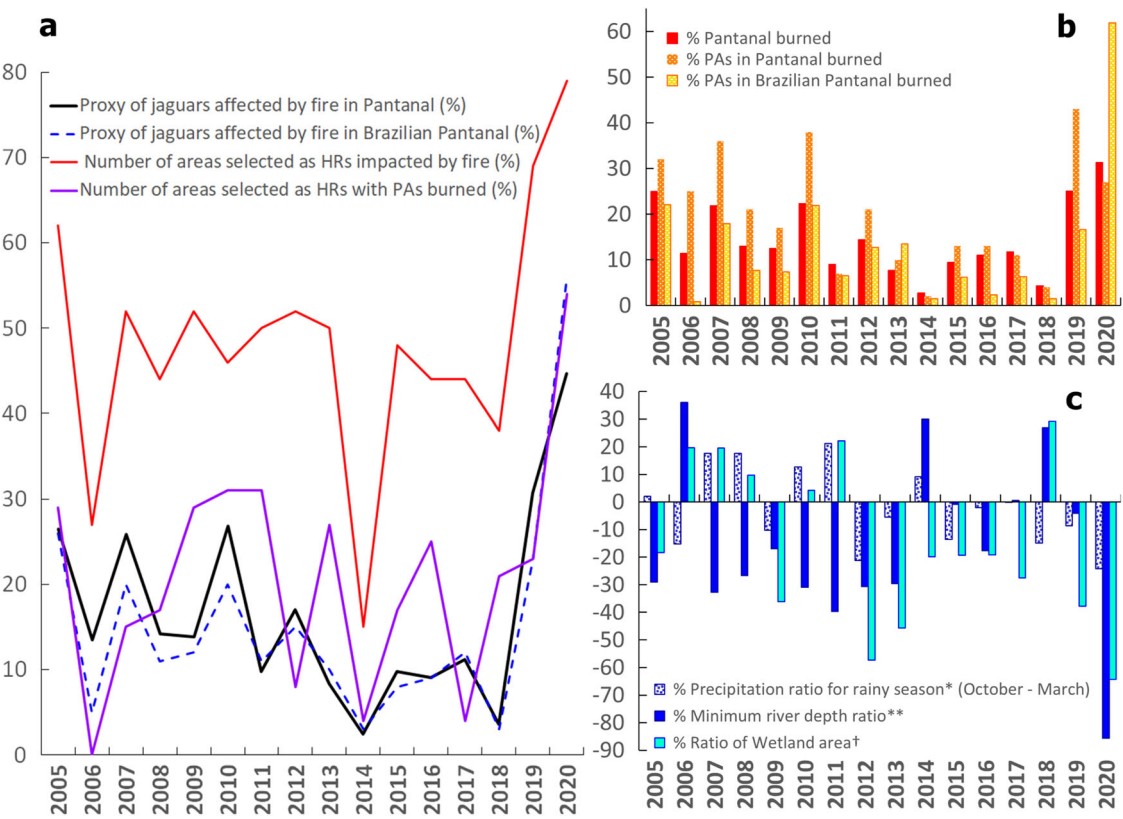

**Fig. 2 Impacts of fire occurrence from 2005 to 2020 on jaguars. a** Percentage of the proxy number of jaguars, home range areas (HRs), and Protected Areas (PAs) used by jaguars affected by fire throughout and within the Brazilian Pantanal. **b** Percentage of the Pantanal and its PAs with fire occurrence. **c** Percentage changes in precipitation and river depth (2005–2020) and wetland flooded areas in the Brazilian Pantanal (MapBiomas 6.0[65]). * % precipitation ratio for the wet season (October–March) (wet season average of monthly medians from 4 stations/average of wet seasons from multiple years (1967–2019). ** % minimum river depth ratio (annual average of minimum river depth from 6 stations/ average (from 6 stations) of the annual medians of minimums (1967-2019). † % ratio Wetland area in the Brazilian Pantanal (Total Wetland annual area/Average Total Wetland area (2005-2019). See SI figures for additional details.

**Impacts of fires on jaguars over the last 16 years**. The occurrence, extent, and intensity of fire differed temporally and spatially, affecting estimates differently over the years (Figs. 1–3 and S10–S15, Table S4). A comparison of the impact of fires on jaguar densities, HRs, and PAs within HRs over the last 16 years showed a notable increase in fire extent and intensity in the last two years (Figs. 1–3, Table 1, SI). However, in contrast to 2020, when the northern Brazilian Pantanal was the most affected area, the 2019 fires affected mostly the Bolivian and areas of the southern Brazilian Pantanal. Nevertheless, the 2020 fire affected an overwhelmingly larger area than in other years (Figs. 1–3, Table 1, SI).

I) A high proxy number of jaguars were also affected by fires in 2019, 2010, 2007, and 2005 (Fig. 2, Table 1), but in all these years, the number of affected individuals was no more than double of the estimated median between 2005–2019.

II) Besides 2020, six years had fire extent and intensity above the historical mean ($\bar{x}$=425 km$^2$, 2005–2019) (Fig. 3). The years 2019 and 2005 had the second and third largest areas affected, with fires reaching 1,196 km$^2$ and 870 km$^2$ within HRs, respectively. These extents were 3.6 and 2.8 higher than the historical median (median=329 km$^2$, 2005–2019), while in 2020, the fire extent on HRs corresponded to 8.3 times the median. Similarly, fire intensity within HRs in 2019 and 2005 was 2 and 2.4 higher than the historical median (median=46 K), while in 2020, fire intensity was >5.

III) In 2011 and 2005, the extent of PAs burned within HRs corresponded to 4.3 and 3.5 times the median (median=115 km$^2$, based on 2005–2019) and doubled the median in 2009, 2013, and

2016 (Fig. 3). Most of the HRs coincided with the protected areas in the Northern Pantanal. Therefore, the largest extent of burned PAs within the HRs matched with the years of higher fire intensity in the northern Pantanal.

Due to the limitations of the original data and sample size, we assumed that the annual estimates of the number of jaguars[36], HRs, and PAs within HRs were the same. Nonetheless, we included in the SI a complementary assessment of the impacts of fire on jaguar home ranges (HRs) that occurred during the monitoring period of jaguar individuals and confirmed that the occurrence of fires within HRs depended on the year and region (Supplementary Note 1, Figs. S10–S15, Table S4). Furthermore, looking at the daily fire occurrence through the years we noticed that although the fire can impact jaguars and the Pantanal along the year, the occurrence and intensity of fires were higher along the dry season, and reached higher values within jaguar HRs (Fig. S13–S15).

## Discussion
Global climate change combined with regional and local anthropic activities suggest an increase in recurrence and extent of wildfires on ecosystems worldwide[31,47,48], affecting in particular regions with higher likelihood of fire occurrences[31] and making natural systems more prone to fire occurrences[21]. Estimates of accumulated burned area in Brazil between 1985–2020 revealed that, among the Brazilian biomes, the Pantanal is the most affected by the fires (with accumulated burned area

Areas selected as Home Ranges by Resident Jaguars (HRs = 48)

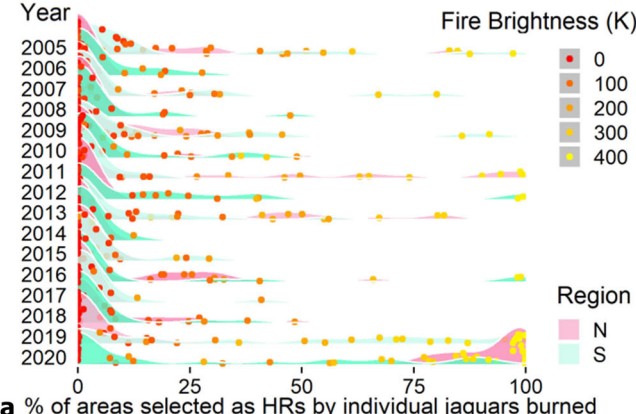

a % of areas selected as HRs by individual jaguars burned

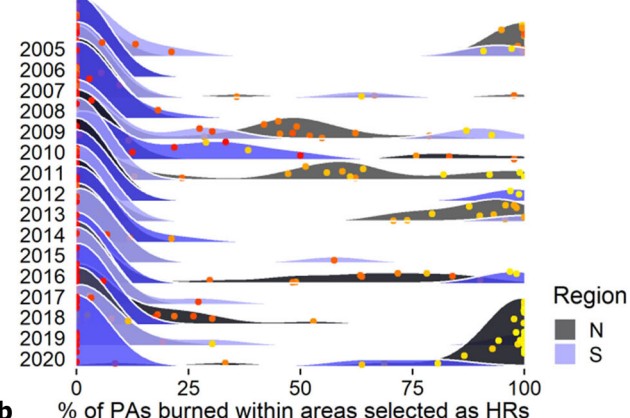

b % of PAs burned within areas selected as HRs

**Fig. 3 Impact of fires on areas selected as HRs in the Pantanal.**
**a** Smoothed frequency distributions of annual percentages of fire occurrence in the Pantanal from 2005 to 2020. Impact of fires on jaguar home ranges and **b** PAs available to jaguars within their HRs. The dots highlight average temperature intensity (fire brightness, in Kelvin) available for each individual for each year.

equivalent to 57.5% of the biome within Brazil)[46]. But 43% of 2020 burned area (≈13% of the Pantanal) had not burned since 2003[19]. Therefore, it is impressive that nearly 1/3 of the Pantanal burned in a single year[17–19] (Figs. 1, 2 and S1, S2). The high number of human-induced fires[17–19,21] combined with the hottest and driest conditions since 1980[17,22,38,49] led 2020 to record the highest daily severity rating (DSR) index of fires for this time period[17,49]. With documented increase of 2 °C in the average temperature[22] and a 40% shortage in rainfall[26,38]. But the fire risk got even higher with the simultaneous occurrence of dry and hot spells, between August and November, when the maximum temperature reached, on average, 6 °C above the normal, accounting for 55% of the burned area of 2020[49].

Most fires started close to the agriculture frontiers[21], but they predominantly affected the natural vegetation (reaching between 91–95% of it in occurrence of fire[50,51] and 96% of it in estimated burned area)[31,46], with tragic consequences for jaguars and the Pantanal biota[17,19,26]. Along with the fires, the severity of the 2020 drought[22,52,53] dropped minimum river depths at around 86% below normal[25,54] (Fig. 2 and S1, S3, S4). Consequently, resulting in several records of animal starvation, dehydration, and death[17,19,26]. And late mortality from indirect causes of fires certainly increased these numbers[26]. Besides, post-fire ecosystem and hydrology changes also had ecological effects with long-term impacts on ecosystem recovery and fire risk[31], impacting resource

quality, availability, and productivity[26,31]. Vegetation productivity declined below −1.5 σ over more than 30% of the natural areas and evaporation decreased (by ~ 9%)[31]. Burned vegetation made the soil more vulnerable to erosion, increasing the runoff (by ~ 5%) over the natural areas[31], and the resulting charcoal and ash contaminated rivers[17].

Many reasons may have contributed to the intensity of the 2020 drought in the Pantanal, from climate[8,22,24,49] to direct and indirect human impacts in the Upper Paraguay River Basin (UPRB)[21,55,56]. In fact, anthropic changes in land use also increased the biome sensitivity to fire-climate extremes)[31]. The shortage of rain throughout the UPRB, particularly in the summer season, is among the main factors, as the basin water balance controls the hydroclimatological dynamics in the Pantanal (Fig. 2 and S3–S9)[22]. The shortage of rain may also be a consequence of increased deforestation in the Amazon rainforest[57,58], as summer rainfall in the Pantanal is strongly associated with the climate of the Amazon[59]. Furthermore, the reduction in wetland flooded areas is historically correlated with the spread of fires (Fig. 2 and S1)[22,28,29]. Low water levels led to the absence of flooding and reduced wetland areas, and the remaining dry vegetation provided flammable material and created favourable conditions for fires to occur[22–24]. In addition, the lack of governmental and human resources and delayed response at federal and local levels[58,60,61] amplified the negative effects of water shortage[17,19,58].

Although historical hydrological series show that extreme drought events occurred in the past[22,25,38,62] (e.g., from the late 1960s to early 1970s, Fig. S3), they also show that the recovery of the Pantanal was conditioned to the subsequent 15 years of regular to exceptional floods (1974 to early 1990s, Figs. S1, S3). Savanna-like vegetation, the predominant vegetation type in the Pantanal, usually recovers from the effects of fires in relatively short periods (months to a few years)[23], depending on the severity and frequency of fires and climate conditions in the subsequent years[23,28,29]. But the resilience of many species may decrease with the annual repetition of extreme fire events[28–30]. Thus, human interventions to prevent (instead to promote) sequential fire events in the same area are paramount[19,23,62,63].

Estimating the effects that uncontrolled extensive fires can cause to the apex predator of the Neotropics in a region considered one of the strongholds for the species can contribute to the conservation of jaguar and other wildlife species, as well as to the debate regarding potential cumulative impact of recurrent wildfires on ecosystems[26,31,51,62,63]. Our results revealed the drastic impact of fire on estimated numbers of jaguars, home ranges, and priority areas for jaguar conservation in the Pantanal was exceptionally high in 2020 and proportionally more severe than the nominal 31% of burned area across the Pantanal (e.g., fires affected 45% of the jaguars and 79% of their HRs). Moreover, the annual comparison showed that 2019 was the second-worst year regarding fire impacts (only behind 2020) and equally extreme compared to historical means[22]. Although the Pantanal is well known for its annual and pluri-annual cycles of wet and dry seasons[7,64], the unusual levels of droughts[22,25,65,66] and fires[17,20,21] in subsequent years are alarming. Furthermore, climate assessment and projections of warmer and dryer conditions for the region in the coming years are equally worrying[22,24,37,38].

We found that 45% of the jaguar population estimated for the Pantanal occupied areas affected by the 2020 fires (Fig. 1). This finding suggests that the fires heavily impacted the jaguars in the Pantanal, even if we assume that the major effects were only temporary displacement. This potential displacement may make it more difficult for jaguars to find new suitable areas, thus increasing territorial disputes and decreasing survival and reproductive success. Furthermore, 2019 ranked as the second-

**Table 1 Proxy for the number of jaguars affected by fires in the Pantanal between 2005 and 2020.**

| | Proxy for the number of jaguars affected by fire in the Pantanal | | | | | Proxy for the number of jaguars in the Pantanal | | | |
|---|---|---|---|---|---|---|---|---|---|
| Year | Ecoregion | Factor effect #Ecoregion/ Median | Brazil | Paraguay | Bolivia | Ecoregion | Brazil | Paraguay | Bolivia |
| 2005 | 441 | 2.0 | 301 | 20 | 120 | 1668 | 1159 | 28 | 481 |
| 2006 | 226 | 1.0 | 58 | 9 | 159 | | | | |
| 2007 | 431 | 1.9 | 232 | 11 | 188 | | | | |
| 2008 | 236 | 1.0 | 127 | 8 | 101 | | | | |
| 2009 | 231 | 1.0 | 139 | 15 | 77 | | | | |
| 2010 | 447 | 2.0 | 232 | 9 | 207 | | | | |
| 2011 | 163 | 0.7 | 127 | 6 | 29 | | | | |
| 2012 | 285 | 1.3 | 174 | 15 | 96 | | | | |
| 2013 | 140 | 0.6 | 116 | 4 | 19 | | | | |
| 2014 | 42 | 0.2 | 35 | 2 | 5 | | | | |
| 2015 | 163 | 0.7 | 93 | 8 | 63 | | | | |
| 2016 | 151 | 0.7 | 104 | 13 | 34 | | | | |
| 2017 | 187 | 0.8 | 139 | 9 | 38 | | | | |
| 2018 | 61 | 0.3 | 35 | 2 | 24 | | | | |
| 2019 | 513 | 2.3 | 267 | 15 | 231 | | | | |
| 2020 | 746 | 3.3 | 649 | 15 | 82 | | | | |
| Median (2005–2009) | 226 | | 127 | 9 | 77 | | | | |

Source: adjusted jaguar density estimates[36] used as a proxy for the number of jaguars in the Pantanal. The Pantanal ecoregion adopted here comprises the legal boundaries of the Brazilian Pantanal biome[119] and the Pantanal Ecoregion[120] within the Upper Paraguay River Basin.

highest year of impact of fire on jaguar population estimates among the 16 years considered (Table 1, Fig. 1). Importantly, we did not consider cumulative impacts on sequential years or fire recurrence in these estimates. Moreover, the available estimates for jaguar abundance we used[36] are very conservative and probably underestimated jaguar populations from the Pantanal by a maximum of 3 jaguars/100 km$^2$. However, the reported density of jaguars may reach up to 12.4 jaguars/100 km$^2$ in northern PAs[5,67,68] and up to 6.5–7 jaguars/100 km$^2$ in the southern Pantanal farms[5,69,70]. Considering that PAs in the northern Pantanal were severely damaged by the 2020 fires, our results show conservative figures for the actual number of jaguars affected by fires.

We used densities estimated from an ecosystem-wide assessment of impacts as a proxy of the proportion of total population reached by fire each year on a regional scale. Fires affected a substantial proportion of estimated individuals in the Pantanal in 2019–2020. In 2020, for instance, 87% of all jaguars affected by fire were in the Brazilian Pantanal. In contrast, the smaller population in the Paraguayan and Bolivian Pantanal had a higher median percentage of individuals affected by fire between 2005–2019. While 45% of jaguars were affected by fire in a single year (2020) in the Pantanal, a study[45] using the same conservative estimates[36] for jaguar abundance in the Brazilian Amazon revealed that 1.8% of the population (1422 individuals) was killed or displaced by fire between 2016–2019. Another report estimated that more than 500 individuals were affected by the 2019 fires in the Brazilian and Bolivian Amazon[71,72]. Based on the same density estimates we found that in the Pantanal — a much smaller biome — more jaguars were affected by fire in single years ($n = 513$ in 2019 and $n = 746$ in 2020). This recent increase in the number of jaguars affected by fire raises a red flag to the supposed stability of the species in the Pantanal, which is currently globally and locally classified as Near Threatened[1,5]. Therefore, we recommend that future assessments by IUCN specialists carefully consider the frequency and intensity of fires as a potentially significant and growing threat to jaguars in the Pantanal, and their effects on long-term populational trends.

Quantifying the occurrence of fire on HRs introduced a functional perspective to understanding the impact of fire on individual jaguars. Similarly, our estimates of the number of affected jaguars revealed a vast amount and extent of affected HRs in the last two years (Figs. 2 and 3). Jaguars are apex predators, often considered as a keystone[73–76] and umbrella species[45,77], highly dependent on large habitat areas[78], dense native vegetation cover[35,79,80], and abundance of prey[67,81]. Considering that jaguars often select areas with high environmental integrity[35,68,78–80], the higher impact of recent fires on HRs corroborates reports showing the increase of natural areas burned in the Pantanal[31,46,50,51]. The proportion of burned forests, for instance, was 10 times higher in 2020 than the estimated median between 1985 and 2019[31]. Sadly, it is likely that much of these burned forests in Northern Pantanal included areas pointed as suitable jaguar habitat and of great interest to the creation of additional PAs[82].

In the Pantanal, HRs are smaller[35,83] and population densities are high[5,67–70] because the biome is a highly productive system[7,55,67], with an abundance of prey species and quality habitat, thus allowing jaguars to meet their spatial needs using smaller areas[35,68,83]. Consequently, floodplain jaguars are also usually larger[44,84]. However, a trend of increasing drought, rising temperatures, and repeated occurrences of exceptional fires would weaken the Pantanal's resilience[22,32]. Importantly to note as well that the occurrence and intensity of fires are frequently higher in the dry season, peaking within jaguars HRs in the years with intense fire occurrence in the Pantanal. This apparent higher impact over jaguar habitat agrees with studies pointing out highest damage in PAs[17,27] (Fig. S20), natural vegetation and particularly in forested areas in 2020[31,46,50,51]. Recurrent impacts may particularly affect the most sensitive species[28–30], resulting in a less productive environment[32], which ultimately decreases the habitat quality of many species. These effects would likely push jaguars to expand their HRs, which would increase disputes for territories and favour a decrease in body size, consequently decreasing reproductive rates and population size.

The extent of protected areas burned is another indicator of how fire can impact biodiversity. Like the HRs, the Pantanal PAs were affected differently in space and time, but the greatest fires occurred in recent years (2019 and 2020). In 2020, fires occurred in 62% of Brazilian PAs — particularly in northern Pantanal — where several portions of PAs overlapping with jaguar HRs were entirely or almost entirely affected by fires (Figs. 1–3). In 2019, however, fires affected the Pantanal PAs in Bolivia, Paraguay and southern Brazil more severely in areas that also overlapped with HRs (Figs. 1–3). Several causes can explain the spread of fires across PAs, including a combination of heat, drought, miscalculated human use of fires, lack of resources and personnel for surveillance and fire control improvement[17–23].

The displacement, injuries, and deaths caused by fire to animals within PAs are worrying because these areas are reportedly richer in diversity and biomass[85,86] (including higher jaguars densities[36,67,87] and are fundamental to safeguarding biodiversity and ensuring the long-term provision of ecosystem services[88,89]. Protected areas are important to jaguars because they provide larger continuous areas of natural dense vegetation cover (such as forests and shrublands), flooded habitats and limit contact with humans, attributes of great influence in jaguar habitat selection[35,78–80,82], and particularly important to females[90,91]. However, although some PAs support up to 12.4 jaguars/100 km$^2$ (e.g., Taiamã Ecological Station - TES)[67], the currently availability of Pantanal PAs alone would not support viable jaguar populations for more than 50 years[87]. Therefore, sustainable management that allows coexistence in private lands is also fundamental for the conservation of jaguars in the Pantanal[5,9–11]. Protected areas of integral protection, such as TES, currently occupy only 5.7% of the Pantanal[7] but were the most affected by fires in absolute area (Fig. S20, Table S5)[27]. The total number of PAs, including the sustainable use ones, corresponds to only 5% of the Brazilian Pantanal (Tables S1–S3)[7,92–96] and around 10% of the entire Pantanal[7], most of it in Bolivia[97]. These percentages are much lower than the minimum of 17% recommended in the Aichi goals for terrestrial ecosystems[7,56]. Furthermore, PAs are also scarce in the Pantanal headwaters (6% of the surrounding Cerrado uplands) (Tables S1–S3, Fig. S19)[7,92–96]. To make matters worse, PAs were reduced by almost 20% in the Brazilian Pantanal in 2007 and have not been expanded in the Cerrado uplands since 2006 (Tables S1–S3, Fig. S19)[93]. The relatively small coverage of protected areas in the Pantanal, which serve as refuges, increases the negative effects of fires, as jaguars are likely displaced into sub-optimal habitats. Consequently, jaguars and other species may struggle to find equally resource-rich sites after being displaced from PAs.

For the long-term survival of the jaguar, it is essential to implement conservation plans that consider the dispersal and reproduction of the species along the Paraguay River[98], increase the network and size of PAs[82], and adequately allocate funding and personnel to maintain the PAs. Furthermore, careful implementation of strategies to mitigate the risk of fire[18,19,62] and other human impacts outside PAs[5–16,89,99] are urgent needs for conservation of the Pantanal. In any case, our results highlight that to sustain viable populations of jaguars and other species, conservation plans for the Pantanal must account for fire impact on PAs and other vital areas for biodiversity.

Although jaguar HRs often overlap with PAs[67,68,87], some individuals may settle in unprotected areas[69,70]. In our sample, we found that 38 HRs partially overlapped with PAs (Fig. 1) and 10 HRs did not. On the other hand, considering the sum of the HR extents and the total area overlapped with the PAs, we found that 20% of the HR extent matched the PAs. Notably, jaguars coexist with different levels of anthropic pressures outside the PAs[4,5,9–16]. Jaguar distribution range has been restricted to 63% of the

Pantanal[5] and even more restricted in the UPRB[100]. Agriculture expansion, particularly cattle ranching and soybean cultivation (Figs. S17, S18)[65], has been identified as the main causes of jaguars' disappearance or decline due to killing and habitat loss[5,9,13].

Sustainable use has been advocated as a conservation strategy in the Pantanal, mainly due to the characteristics of the region, where cattle ranching uses as pastures the natural areas restricted by the Pantanal flooding regime since the 17th century[7,23]. In recent years, ecotourism has also gained great importance[55,101,102]. However, there are risks in relying on sustainable use as a core strategy for 90% of the biome (95% of Brazilian Pantanal), and exposure to human-induced fires is one of them[21,31].

Fire is a fundamental factor acting on the dynamics of the Pantanal vegetation[23,28,29]. However, repeated uncontrolled fires can drastically impact forests and other habitats critical to the jaguars and increase the area for cattle ranching, therefore increasing the risk of livestock depredation and retaliatory hunting[11]. Thus, the conservation of the jaguar and other animal species in the Pantanal is critically linked to fire management and the use of private lands because the increased fire may extend and aggravate other anthropic impacts (Fig. 4). This work highlights the significant increase in the extent and severity of recent fires in the Pantanal and how these fires have affected jaguars. Further studies that estimate natural habitat recovery and fire recurrence and assess real-time and long-term effects of fire on jaguars and other species are critical to guide fire management and conservation.

Changes in the climate[8,22,24,37,38], landscape and water use in the UPRB over the last four decades[7,18,56,65] are cumulative threats that may interfere with water recharge and vegetation resilience in the Pantanal. Global temperatures may increase up to 1.5 °C over the next five years[37], in addition to the 2 °C already recorded since 1980. By the end of the 21st century, scientists estimate increases of 5 − 7 °C in the temperature and the frequency of climatic extremes and a 30% reduction in average rainfall[8,37,38]. Until 2019, pastures covered 15.5% of the Brazilian Pantanal and agriculture about 0.14%[25]. However, agriculture and pastures occupied 60–65% of the surrounding Cerrado uplands within the UPRB[7,55,56], an occupation similar to the adjacent Paraguayan Chaco and Bolivian Chiquitano Forest[7,103,104]. And future projections estimate a loss of 14,005 km$^2$ of native vegetation from 2018 through 2050[105]. Consequently, this land occupation impacted the main headwaters of the Pantanal rivers and ultimately the entire Pantanal[6,56,106,107]. Furthermore, by 2019, 47 hydroelectric power plants were installed or in operation, and another 133 were planned, totalling about 180 potential dam projects in the Brazilian UPRB[108]. Besides, most of these projected hydropower infrastructures will overlap with the distribution of jaguars, also in the adjacent biomes, impacting negatively the species particularly in Brazil[15]. These economic and infrastructure activities in the surrounding highlands frequently ignore their cumulative impacts[109] and affect the Pantanal in different ways (Fig. 4, S17, S18), including its drainage dynamics and flood pulses, with consequent impacts on drought duration and fire spread[17,19,22–24](Figs. 1–4, SI). This combination of factors probably intensifies the Pantanal droughts, particularly the periodic sequence of dry years.

Therefore, a critical point is how human actions can exacerbate such extreme events[7,21,31,55,106,110] and make fire control even more difficult[19,23,62] or, on the opposite, contribute to minimize the overall impacts of drought and fires and promote biodiversity conservation[19,63] (Fig. 4). Given that the rainfall remained below average in the last wet seasons[53] (Figs. S1, S3–S8) and that a severe drought persisted in 2021[111], a surveillance protocol for rapid response and programs for fire management, mitigation of human impacts and ecosystem recovery are needed[19,23,62,63]. If

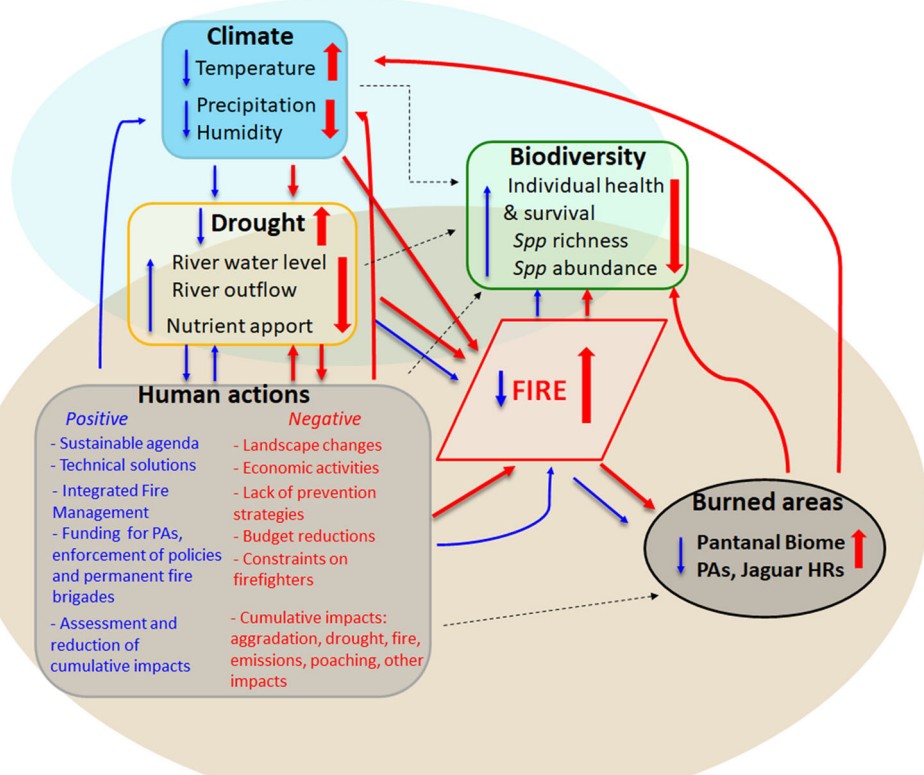

**Fig. 4 Scheme summarizing the main impacts of fires in the Pantanal.** The red arrows are intentionally larger and show a feedback loop linking increased negative human impacts, climate change, and drought to increased fires and burned areas, with a consequent negative impact on biodiversity. The blue arrows describe a feedback loop for fire control and impact mitigation. The dashed arrows denote other relevant effects in the biome (e.g., cumulative effects from infrastructure such as hydroelectric power plants, river waterways, water and soil pollution from legal and illegal mining and agriculture, poaching and illegal wildlife trade, opportunistic exploitation of burned areas, as well as natural climate constraints.

such measures keep lacking, a tragedy similar to the 2020 fires may be repeated in the coming years (Fig. 4). And Pantanal native vegetation may be reduced to only about 62% by 2030[21]. To make matters worse, the government budget allocated for fire control and firefighting for 2021 was reduced to 65.5% of the 2019 budget[61] and all funds allocated to the environment were reduced to the lowest level in 20 years[61,112], with serious complaints of misuse[113], embezzlement[114] and wood-smuggling probe[115].

The extent of the recent wildfire in the Pantanal has signalled that fire is a potential threat to the long-term conservation of the jaguar. Furthermore, fires severely affected other species and human activities[17,19,23], demanding an immediate mitigation plan[18,19,62]. In fact, permanent fire brigades have been established, and an animal rescue centre is under construction in response to the effects of the recent extensive fires in the Pantanal. Although actions are underway at local levels, the warming and drying trend[22,24,37,38] is also a combination of global warming[8,37] and rapid land-use changes[7,18,65] (Figs. S17, S18), with cumulative impacts in the UPRB and Pantanal wetlands (Fig. 4). Therefore, the immediate reduction of deforestation in the Amazon and Pantanal and the establishment of a forest restoration plan in the UPRB are critical. The lack of sufficient mitigatory actions may throw the Pantanal into a perverse vortex (increasing feedback of cumulative negative impacts, (Fig. 4), thus affecting the survival of jaguars and the various species under their umbrella, as well as human welfare.

## Methods
**Study area.** The Pantanal is the largest wetland in the world[7,17] and is characterized by a mosaic landscape with floodable and non-floodable areas containing grasslands, forests, open woodlands, and temporary or permanent aquatic habitats[6,7,65,79]. The Pantanal wetland is located within the Upper Paraguay River Basin (UPRB), which comprises a drainage area of 600,000 km² (362,380 km² in Brazil)[108,116,117]. The Pantanal is about 160,000–179,300 km² distributed across Brazil (78–85%), Bolivia (15–18%), and Paraguay (1–4%)[7,118–120]. The UPRB contains the river springs that drain into lowlands and floods the Pantanal[7,56], which stores this water and delivers it slowly westward to the Paraguay River[7,38]. The wet and dry seasons are well-defined, with most annual rainfall falling from November to March and defining a seasonal flood pulse that controls and shapes the biota in the channel-plain system[64,121]. Leading to a flood season which predominates between November and March in the north and between May and August in the south[22]. In turn, seasonal floods impact nutrient cycling, vegetation, primary productivity, and wildlife[117]. In addition to flooding, fire is another element that interferes with species abundance and composition[28,29]. While small amounts of fire may promote diversity, the recurrence of high-intensity fires is more likely to be detrimental[23,28–32].

Precipitation, and temperature differ temporally and spatially in the Pantanal wetlands and the UPRB[8,121–125]. According to the Köppen classification[124], the UPRB and Pantanal include mainly tropical zones with dry winters (Aw) and annual average precipitation around 1400 mm. The UPRB also includes a tropical monsoon (Am) region with rainfall between 1300 and 1600 mm, a small tropical rainforest (Af) in the south with rainfall between 1400 and 1800 mm, and an even smaller region classified as a humid subtropical zone (Cfa)[124]. Rainfall is usually higher in northern-northeastern (2000 mm) and southern (1800 mm) areas, coinciding with the uplands (plateaus)[121,124]. In central Pantanal, rainfall is lower, with about 900 mm (and 800 mm near the Bolivian Chaco)[121,124,125]. The Pantanal is bordered by the savanna or Cerrado to the east (which covers the surrounding plateaus), the Amazon to the north, the Atlantic Forest to the southeast (represented by semi-deciduous and deciduous forests), and the Chaco to the southwest. These neighbouring biomes biogeographically influence the Pantanal's biodiversity.

**Fire, precipitation, river depth and GIS boundary data.** We used Google Earth Engine (GEE)[126] to obtain near-real-time (NRT) active fire locations in a rasterized form (1 km resolution) with one or more fire occurrences per pixel[42]. These data were processed by the Land, Atmosphere Near real-time Capability for EOS

(LANCE)/ Fire Information for Resource Management System (FIRMS) using the standard MODIS MOD14/MYD14 Fire and Thermal Anomalies product[42,43]. We used fire data from January 2005 to December 31, 2020 in the main analyses. This period corresponded to the jaguar monitoring time (2005–2016), but we also evaluated fire impacts in subsequent years. We used both the occurrence of fires and their intensity (temperature in Kelvin) and adopted a threshold of 325 Kelvin as a determinant of fire occurrence[42,127]. Therefore, we assumed the occurrence only in pixels with fire intensity above this value.

As a spatial limit of the Pantanal, we adopted a merged image of the legal boundaries of the Brazilian Pantanal biome[119] and Pantanal Ecoregion[120] within the UPRB, totaling 160,426 km$^2$. We calculated fire occurrences separately within each country's boundaries[128]. The Pantanal area within Brazil corresponded to 150,893 km$^2$ (150,355 km$^2$ of the legal biome[119] merged with additional Pantanal ecoregion[120] areas within Brazil). The Pantanal ecoregion[120] corresponded to 26,399 km$^2$ within Bolivia and 1970 km$^2$ within Paraguay. Vectors for countries, ecoregion, and PAs boundaries were rasterized and resampled to match the 1 km resolution and then reclassified using GEE[97,126] and the raster package[129] from R statistical software[130]. Estimates of annual land-use changes and wetland extent were based on MapBiomas collection 5.0[65] and complemented with data on rainfall[52,53] and river water levels[25]. The polygons of protected areas were downloaded from GEE[97,126] and Brazilian Ministry of Environment geodatabase[92,95]. Some private protected areas may be missing because data were unavailable[7]. We supplemented our discussion using complementary information on PAs[93–96], estimates of fire impact[17,27], relationships between fires and precipitation, river water levels, land-use change, and wetland extent, among other data (SI).

**Statistics and reproducibility**. We evaluated the impact of fire in the Pantanal by overlapping raster images of the annual occurrence of fires and the Pantanal extent within each country. We reclassified the Pantanal boundaries so that the sum of the cell values was 1 and then multiplied these values by the raster of fire occurrence. This multiplication resulted in a distribution of the occurrence of fires, with the sum of these cells corresponding to an estimated proportion of the impact of fire in the Pantanal of each country. The mean (or median) annual fire intensity was calculated based on the pixels' mean (or median) values.

A similar process of resampling and reclassifying raster images was applied to evaluate the impact of fire on the PAs of the Pantanal. First, we calculated the extent of PAs in the Pantanal. Second, we calculated the extent of PAs impacted by fires — i.e., the probability of fire occurrence per pixel based on the multiplication of the Pantanal PAs raster by the fire occurrence raster. Then, we calculated the ratio between the PAs impacted by fire and the total extent of the Pantanal PAs in each country.

In order to evaluate the congruence with other fire datasets[20,46], in relation to the impacts of yearly fires on the Pantanal, we performed a Spearman correlation analysis (Fig. S2).

**Proxy for the number of jaguars affected by fires in the Pantanal**. We used estimates of the population density of jaguars[36] occurring in the pixels reached by fires[42] as a proxy for the number of jaguars affected (e.g., potentially displaced, injured, or killed by fires[45]) in 2020 and the previous 15 years. We used these population density estimates in a similar way to a study evaluating a proxy for the number of jaguars displaced in burned areas in the Amazon[45].

In the study by Jędrzejewski et al.[36] population estimates were derived from 80 studies of camera traps spread across the jaguar distribution between 2002 to 2014. Population density and probability of occurrence were then modelled as response variables to environmental covariates, such as net primary productivity[36]. Finally, to adjust the estimates to the actual jaguar range[1], the authors multiplied the population density estimate by the probability of occurrence estimate[36].

We thus clipped the raster image output from Jędrzejewski et al.[36] containing jaguar abundance estimates with the Pantanal polygon masks of each country and adjusted the resolution to 1 km. As the original information corresponded to the estimated number of jaguars per 100 km$^2$, we converted this information to a 1 km resolution by dividing the cells by 100, thus obtaining the number of jaguars per 1 km$^2$. Therefore, the sum of the pixels corresponded to a proxy for the total number of jaguars within the boundary of the Pantanal area to be assessed (for Brazil, Bolivia, Paraguay, or the entire Pantanal).

Next, we selected the pixels of jaguar density estimates overlapping with the occurrence of fire. Thus, the sum of the pixels with fire records corresponded to a proxy for the estimated number of individuals impacted by fire in the Pantanal in each country. Finally, we calculated the correspondent percentages of jaguars impacted by the fire.

Importantly, the estimates by Jędrzejewski et al.[36] were conservative (with a limited number of study sites in the Pantanal), and their model favoured forested regions. Moreover, these authors did not explicitly consider other important factors that may affect abundance in the Pantanal, such as prey density[67,68]. Despite these shortcomings, the map by Jędrzejewski et al.[36] is still the best available proxy to point the number and spatial variations of jaguars, and it has already been successfully used for comparisons and estimates of fire impacts on jaguars from the Amazon[45].

**Jaguar home range estimates**. We used published data[44] to estimate jaguar home ranges and evaluate the impact of fire on home ranges (HRs) during 2020 and the previous 15 years. We gathered GPS data on the movement of 56 individual jaguars tracked at seven sites[36] in the Brazilian, Paraguayan, and Bolivian Pantanal. From these data, we used 48 individuals classified as residents. We excluded individuals with insufficient data or classified as non-residents (Figs. S11, S16). Individual residency status was evaluated by analysing the asymptotic behaviour of semi-variograms (Fig. S21, Table S6)[35,68,83] and complementary statistics, such as the estimated number of range crossings ($N_{area}$ or $DOF_{area}$), with the continuous-time time movement modelling (ctmm) R package[40,131,132]. Individuals were classified as residents if they inhabited the home-range area during the monitoring period, had $DOF_{area} > 5$[133], or obtained an asymptote in their semi-variogram[35,68,83] (Fig. S21, Table S6). The minimum sampling period used was 27 days, and the maximum was 591 days (SI).

Data cleaning and preparation for temporal order and duplicates were performed in R[130], using amt[134] and ctmm[131,132] packages. We calculated individual jaguar home ranges as indicative of areas selected as home ranges (HRs) using the Autocorrelated Kernel Density Estimator (AKDE), from the ctmm R package[131,132], and the same grid alignment and resolution as the fire raster images. From each AKDE, we calculated the probability mass function, an indicator of the intensity of jaguar space use within the AKDE-derived raster images, and multiplied this value by the raster images of fire occurrence. The sum of the resulting probabilities at each pixel meant the proportion of individual jaguar HRs impacted by the fire. The annual fire intensities within HRs were calculated by averaging the fire intensity values recorded at each pixel. Lastly, we calculated the frequency distribution of jaguars in PAs, i.e., the extent of HRs included in PAs. Then we estimated the extent of HRs containing PAs with fire occurrence. To do so, we first multiplied the estimated probability mass function of each jaguar (corresponding to the jaguars' AKDE) by the occurrence of PAs. Next, we multiplied these two layers by the raster images of fire occurrence. These analyses (Figs. 2, 3) consisted of comparing the impact of fire in all HRs ($n = 48$) over time (2005–2020).

We performed an analysis of variance (ANOVA) to understand the effect of year and region of fire occurrence on jaguar HRs and compared models including each variable alone, additive and interaction effects (Table S4. Furthermore, we also considered the percentage of fire occurrence matching individual jaguar HRs areas only during the GPS monitoring period and compared these analyses with those projected for many years (Figs. S10–S12, Table S4). Finally, we also look at the daily intensity of occurrence of fire through the year for the Pantanal and for the areas selected as HRs (Figs. S13, S14), as well as coinciding with HRs during the real time monitoring period (Fig. S15).

**Reporting summary**. Further information on research design is available in the Nature Research Reporting Summary linked to this article.

## Data availability

Original jaguar data[44] associated with this publication are available at https://doi.org/10.5061/dryad.2dh0223 (Dryad Digital Repository). We provided raw and processed data at https://doi.org/10.6084/m9.figshare.17698595.v1.

## Code availability

We followed J. Fieberg and J. Signer[134] scripts for cleaning and preparing the basic movement data. The analyses of the jaguar home range followed[131,132]. First, we ran AKDE Home Ranges in R. Then, we entered and merged all the HRs in GEE and reran AKDE in a common grid with the fire raster output.

A) Google Earth Engine example from 2020 (Main Code) https://code.earthengine.google.com/f0ae619db0404f606562d33290416277.

The same code was applied filtering other years (2001 to 2019).

B) R scripts with raster operations accounting for fire impacts on areas, jaguar abundances and home ranges are available at https://doi.org/10.6084/m9.figshare.17698595.v1.

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

## Acknowledgements

This research was supported by the Coordenação de Aperfeiçoamento de Pessoal de Nível Superior – Brasil (CAPES) – Finance Code 001 and Fundação de Amparo à Pesquisa do Estado de São Paulo - FAPESP 2018/24891-5. We thank the Committee members (Andrea Larissa Boesing, Leandro Reverberi Tambosi, and Eduardo Martins Venticinque) who encouraged us to proceed with the paper idea. We also thank Marcus Suassuna Santos (SGB/CPRM) for his help in accessing hydrological data.

## Author contributions

A.E.B., P.I.P., R.G.M., and C.H.F. conceptualized the analysis, A.E.B. undertook the analysis, A.E.B., P.I.P., R.G.M. led the writing of the manuscript. All the other co-authors (R.P., L.G.R.O.S., W.M.T., D.Z.K., F.R.T., C.E.F., F.C.C.A., J.J.T.) provided text contributions and/or figure feedback. R.G.M., D.Z.K., and C.E.F. also provided photos with authorized usage.

## Competing interests

The authors declare no competing interests.
