## [Peer Review File · Communications Biology]

Reviewers' comments:

Reviewer #1 (Remarks to the Author):

Dear Author(s),

The manuscript, as written, is prepared well. It was enjoyable to read and for the most part, flowed nicely and logically. The following are suggestions to increase flow for the reader and bolster some sections that would benefit from additional reference and discussion.

1. The main argument presented in the manuscript is that recent, large fires have had a disproportionate, negative impact on jaguar home ranges in the Pantanal. The data, figures, and discussion support this conclusion. Where the manuscript fell short for me was discussing why this should be of major concern. Fire data used in this paper range from 2005-2020. Were the extreme fires detected in 2019 and 2020 an anomaly? I would like to see, if possible, incorporation of fire data pre-2005 and projections of fire intensity in future years (e.g., 2030, etc).
2. These data exist for climate and land-use change as well, which should also be more thoroughly discussed. In your Abstract (Line 32) you list "climate and land-use change" are threats to the jaguars' long-term survival but this isn't discussed in further detail in your Introduction or Discussion. See Guerra et al. 2020 (Drivers and projections of vegetation loss in the Pantanal and surrounding ecosystems | <https://doi.org/10.1016/j.landusepol.2019.104388>) and Alho et al. 2019 (Threats to the biodiversity of the Brazilian Pantanal due to land use and occupation | <https://doi.org/10.1590/1809-4422asoc201701891vu2019L3AO>) among others. If climate, land-use, and fire intensity are all showing upward trends historically, and projected, this would be a worthy discussion/addition to your manuscript.
3. Line 77 – word choice "countless" ↓ where possible, avoid ambiguities. You may not have an exact count but the word choice here suggests there were too many jaguars injured/killed to count when on Lines 125-126 the fires impacted an estimated 45% jaguars (n = 746 individuals).
4. Lines 80-85 discusses fire's long-term negative impact on GPP. Caution, other sources have found increases in GPP following massive fire disturbance. Systems, disturbances, and impacts are not static: for the Amazon, see Rap et al. 2015 (Fires increase Amazon forest productivity through increases in diffuse radiation | <https://doi.org/10.1002/2015GL063719>) and for another example, Ryu et al. 2018 (Satellite-Based Evaluation of the Post-Fire Recovery Process from the Worst Forest Fire Case in South Korea | <https://doi.org/10.3390/rs10060918>).
5. Lines 92-103 is a discussion of data sources and processing, these would be better suited under the Methods section.
6. Line 181 – the results presented support drastic impact to home ranges (as primary impact of fires) and then to jaguar displacement, injury, death (as secondary impacts). See Comment 1. above.
7. Lines 182-183 conflict with Line 194. On Lines 182-183 the manuscript states "The impact of fire on population size, home ranges, and priority areas for jaguar conservation in the Pantanal was exceptionally high in 2020 and proportionately more severe..." But Lin 194 states "...even though the major effects were only temporary displacement." Followed by a potential factors such as crowded home ranges and decreased reproduction. Unless you specifically plan to discuss the fire's direct impact to population size, I would consider rewording Lines 182-183.
8. Lines 189-191 mention increasing concern over climate projections. See Point 2. – it is mentioned in the Abstract and again briefly as one sentence here. Consider elaborating with specifics of what those projections mentioned for the Pantanal and what those impacts mean to jaguar home ranges. Is

it fragmentation, reduction in core habitat size, genetic isolation, loss of localized populations, reduced prey populations, poor habitat quality, drier/wetter, more severe weather events and disturbances, all of the above? More details in the form of sources/references/supporting data that back your claims would be welcomed.

9. Lines 207-208 "We noticed astonishingly absolute numbers of individuals..." I'm not exactly sure what this means. Word choice "astonishingly" consider replacing with something less sensational. "...absolute numbers" conflicts with "countless individuals" (Line 77).

10. Line 217 – you provide estimates for numbers of impacted jaguars, see Points 3., 7., and 9.

11. Line 230 word choice – consider replaced "...had not been burnt..." with "...had not burned..."

12. Lines 270-277 – Do you have a source that shows support for jaguar selection/preference for habitat within protected areas. I may have missed it in your manuscript but further discussion here would help support disproportionate impact of fires in PAs.

13. Lines 287-292 How many human-caused fires have been documented in recent years? Is this trend congruent with intensity of fires?

14. What was responsible for the 2019 and 2020 fires? Did these the fires occur in a relatively short window of time e.g., not allowing animals the opportunity to move out of burning areas or were they sporadic throughout the year? Or stated more simply, how many 'major fire events' (would need to be defined) occurred in the worst years. Line 64 mentions fire outbreaks have increased 400%, and Lines 246-248 mention discuss possible causes of fire spread but what was the likely origin cause of the fires?

15. Line 284 introduces the first use of the abbreviation UPRB but this isn't defined until Line 328. Remove explanation of abbreviation from Line 340.

16. Lines 351-353 can be moved to Introduction.

17. Figures were self-containing and readable with exception of Fig. 2. Consider replacing "Number of areas selected as HRs with PAs burned (%)" with another color or line choice. Orange is too close to red in printed manuscript.

Reviewer #3 (Remarks to the Author):

The authors examined how changes in the frequency and intensity of wildfires in the Pantanal ecoregion have impacted jaguars. Using remote sensing data, jaguar population density estimates and GPS data, the authors show that the most recent fires of 2020 were the most intense and impacting jaguar populations, their home ranges and the protected areas within the jaguar home ranges. The authors achieve a lot in this paper; not only characterising the fire history over a long time period and the impacts of fire for people and biodiversity, but also characterises the impacts of these fires on jaguars. Based on my knowledge, this is the first work of its kind within the Pantanal ecoregion. I believe this paper will be of interest to range of ecologists, particularly those interested in wildfire impacts.

There is a lot of additional work and analyses performed by the authors that didn't make it to the main text (e.g., as exploring relationships between fires and precipitation, river water levels, land-use change, and wetland extent), and I wonder whether it would be best to split the paper. One paper that focus on the characterisation of fires over time, and the implications for conservation, and a

second paper focusing only on fire impacts on jaguars. Alternatively, a journal with a longer format that allows for the complete discussion of implications might also be a good option. I am not saying the work is not publishable, I think the authors have done some great work here, I just wonder if the current format and structure is the best option.

Ln 61 - 62: Here the authors mention that extent of the Pantanal that was impacted by the 2020 fires compared to the previous years. It was not clear if this a result of your work, or based on previous work. In results (ln 115), the same numbers are mention, so I assume these are results from your study. If this is the case, I am not sure I would mention these values in the introduction to avoid duplication and to avoid starting with your results.

The authors estimated home ranges for 48 resident jaguars monitored between 2005 and 2016, and then compared the extent of the home ranges that the fires burnt for each year. It wasn't clear to me how the authors estimated the impacts of the fires on home ranges from 2017 to 2020, without GPS data from those years. Did they assume the 48 residents maintained their ranges for those years?

Fig. 1: If the authors have not already done so, I would recommend they check if this figure is colour-blind friendly due to the inclusion of red and green.

REVIEWERS' COMMENTS:

Reviewer #1

Remarks to the Author:

The manuscript, as written, is prepared well. It was enjoyable to read and for the most part, flowed nicely and logically. The following are suggestions to increase flow for the reader and bolster some sections that would benefit from additional reference and discussion.

We thank the reviewer for their detailed comments. In tackling them many of the page numbers have shifted slightly and so to avoid to confusion, all page numbers listed in our responses correspond to the new page numbers in our revised document.

1. The main argument presented in the manuscript is that recent, large fires have had a disproportionate, negative impact on jaguar home ranges in the Pantanal. The data, figures, and discussion support this conclusion. Where the manuscript fell short for me was discussing why this should be of major concern. Fire data used in this paper range from 2005-2020. Were the extreme fires detected in 2019 and 2020 an anomaly? I would like to see, if possible, incorporation of fire data pre-2005 and projections of fire intensity in future years (e.g., 2030, etc).

Based on our perception of research gape, we decided to focus on the analysis of impacts of fires on jaguars since our first version. We believe that this would be the best approach considering some restrictions of space, as well as for sake of clarity in the research article. The period constraint we adopted was mostly based on the availability of the jaguar movement data (available from 2005 and 2016), although the fire data we used was available from year 2000. Thus, we based our analyses mostly on the time lag of the jaguar movement data, but we also decided to extend the analysis to 2020 to be able to include an assessment of 2020 megafires impact. Therefore, although we had gathered some additional data and done some extra analysis, in our attempts to shrink the text on the earlier version submitted, we removed, or moved to the SI, parts of text that could address at least in part this topic. Therefore, we return part of it to the main text, as well as we included updated references.

We understand that the main concern of the reviewer is regarding the extension and intensity of fire in the previous years (before 2005), as well as possible projections of fire in future years. The absence of fire data is a problem here. The dataset we used started in the year 2000 (FIRMS 2020). And the majority of other sources we checked, that are based on the MODIS satellite data, are also available from around this year. On top of that, some new datasets delimitating fire scars became recently available since 1985 derived from the LANDSAT satellites data, but only to the Brazilian Pantanal (MapBiomass 2021). As for fire predictive models, these are often compared against past data to be able to evaluate model accuracy (Jolly et al. 2015 <https://www.nature.com/articles/ncomms8537>), but we look after the fire trends and included information of some relevant papers in the discussion (e.g. Libonati et al.

2022, <https://iopscience.iop.org/article/10.1088/1748-9326/ac462e>). We also included a correlation comparison of the fire data we used with other two datasets (SI_Fig.S1a, SI_Fig.S1b), brought back some relevant information from the SI and addressed the related topics in appropriated parts of the discussion (lines 194-249 and 298-302 in the version using track changes).

2. These data exist for climate and land-use change as well, which should also be more thoroughly discussed. In your Abstract (Line 32) you list “climate and land-use change” are threats to the jaguars’ long-term survival but this isn’t discussed in further detail in your Introduction or Discussion. See Guerra et al. 2020 (Drivers and projections of vegetation loss in the Pantanal and surrounding ecosystems | <https://doi.org/10.1016/j.landusepol.2019.104388>) and Alho et al. 2019 (Threats to the biodiversity of the Brazilian Pantanal due to land use and occupation | <https://doi.org/10.1590/1809-4422asoc201701891vu2019L3AO>) among others. If climate, land-use, and fire intensity are all showing upward trends historically, and projected, this would be a worthy discussion/addition to your manuscript. We fully agree with the reviewer. We included some new research papers (e.g Marques et al. 2021, Kumar et al. 2022, Mataveli et al. 2020) and brought back some other relevant paper we had removed or left in the SI (including the ones mentioned by the reviewer), to address and improve this topic in appropriate parts of the discussion (along with discussion of fires in lines 194-242 and considering other cumulative impacts in the Pantanal lines 385-417 and in the SI).

3. Line 77 – word choice “countless” where possible, avoid ambiguities. You may not have an exact count but the word choice here suggests there were too many jaguars injured/killed to count when on Lines 125-126 the fires impacted an estimated 45% jaguars (n = 746 individuals).

We addressed this in the (line number 80) changed the word “countless” to “several”.

4. Lines 80-85 discusses fire’s long-term negative impact on GPP. Caution, other sources have found increases in GPP following massive fire disturbance. Systems, disturbances, and impacts are not static: for the Amazon, see Rap et al. 2015 (Fires increase Amazon forest productivity through increases in diffuse radiation | <https://doi.org/10.1002/2015GL063719>) and for another example, Ryu et al. 2018 (Satellite-Based Evaluation of the Post-Fire Recovery Process from the Worst Forest Fire Case in South Korea | <https://doi.org/10.3390/rs10060918>).

We rewrite this paragraph and included new references, including the suggested ones (lines 83 to 86, Rap et al. 2015, Ryu et al. 2018 and Kumar et al. 2022).

5. Lines 92-103 96 - 107 is a discussion of data sources and processing, these would be better suited under the Methods section.

These methodological points are indeed addressed in detail in the methods. However, considering that in this writing structure the methods come last (and consequently will be read only after the discussion by the reader), we believe that both, the brief mention to the origin of the datasets, and some theory backing up our objectives can benefit readers’ understanding. For this reason, we respectfully decided to maintain the paragraph in place. We also modified and included some additional explanation in response to comment 20 in this part.

6. Line 181 – the results presented support drastic impact to home ranges (as primary impact of fires) and then to jaguar displacement, injury, death (as secondary impacts). See Comment 1. above.

We unfortunately did not have jaguar movement data with a broader temporal range but we linked these statements with the broader discussion responding to Comment

1 (lines 194-249). We also explored temporal changes in the intensity of fire occurrence through the year and how and when it may have caused higher impacts on jaguars HRs and to overall Pantanal areas (see additional graphs in the SI_Figs 3d, 3e, 3f, lines 182-190 in the results and lines 297-302, 309-313, 330-336 in the discussion). We included one of these figures (the SI_Fig.3d) among the 10 extended data that can go with the main paper in the digital version.

7. Lines 182-183 conflict with Line 194. On Lines 182-183 the manuscript states “The impact of fire on population size, home ranges, and priority areas for jaguar conservation in the Pantanal was exceptionally high in 2020 and proportionately more severe...” But Lin 194 states “...even though the major effects were only temporary displacement.” Followed by a potential factors such as crowded home ranges and decreased reproduction. Unless you specifically plan to discuss the fire’s direct impact to population size, I would consider rewording Lines 182-183.

We rewrote this part changing “population size” to “estimated numbers of jaguars” to clarify (line numbers 251-252).

8. Lines 189-191 mention increasing concern over climate projections. See Point 2. – it is mentioned in the Abstract and again briefly as one sentence here. Consider elaborating with specifics of what those projections mentioned for the Pantanal and what those impacts mean to jaguar home ranges. Is it fragmentation, reduction in core habitat size, genetic isolation, loss of localized populations, reduced prey populations, poor habitat quality, drier/wetter, more severe weather events and disturbances, all of the above? More details in the form of sources/references/supporting data that back your claims would be welcomed.

We worked to improve this topic in the discussion and added references (Lines 194-252, 385-416, 297-302, 309-313, 330-336; please see also the responses to point 1, 2 and 6).

9. Lines 207-208 “We noticed astonishingly absolute numbers of individuals...” I’m not exactly sure what this means. Word choice “astonishingly” consider replacing with something less sensational. “...absolute numbers” conflicts with “countless individuals” (Line 77).

We adjusted these in the lines 276 - 279 to “Fires affected a *substantial proportion*” of estimated individuals in the Pantanal in 2019–2020. This adjective corresponds to the 45% of estimated individuals affected by fire in 2020, mentioned in the anterior paragraph. We used “*substantial proportion*” to avoid repetition of the 45% value when we needed to mention the same information.

We changed “*countless*” in the line 80 to “*several*”.

10. Line 217 – you provide estimates for numbers of impacted jaguars, see Points 3., 7., and 9.

We rewrite these previous points to make them more coherent.

11. Line 230 word choice – consider replaced “...had not been burnt...” with “...had not burned...”

We replaced as suggested but moved this part to the beginning of the discussion (line numbers 200-201).

12. Lines 270-277 – Do you have a source that shows support for jaguar selection/preference for habitat within protected areas. I may have missed it in your manuscript but further discussion here would help support disproportionate impact of fires in PAs.

We elaborated on the topic linking it to the high amount of natural dense vegetation cover and flooded areas in PAs and added references (Morato et al 2018, Alvarenga et al. 2021 among others) pointing these as important resources in jaguar habitat selection (lines 330-336, but also 297-302, 309-313).

13. Lines 287-292 How many human-caused fires have been documented in recent years? Is this trend congruent with intensity of fires?

We included new references supporting this trend e.g Marques et al. 2021²¹, Kumar et al. 2022³¹ among others (lines 194-249, 297-300, 385-417). Some of the take home messages of these papers are: that human activity and consequent change in land use increased the risk of fire in the Pantanal, making natural systems more prone to fire occurrences during extreme droughts³¹, and that based on the impact of these changes, in 2020 for instance, 76% of the registered fire foci occurred in regions classified as high risk of fire²¹. Furthermore, if the burning areas observed in 2020 are recurrent, about 20% of the Pantanal will be for agricultural and pasture use and only about 62% will correspond to native vegetation.

14. What was responsible for the 2019 and 2020 fires? Did these the fires occur in a relatively short window of time e.g., not allowing animals the opportunity to move out of burning areas or were they sporadic throughout the year? Or stated more simply, how many ‘major fire events’ (would need to be defined) occurred in the worst years. Line 64 mentions fire outbreaks have increased 400%, and Lines 246-248 mention discuss possible causes of fire spread but what was the likely origin cause of the fires?

We discussed better this topic based on references and looking in detail to fire data as we mentioned in response to point 1 and others. We also explored some additional graphs (see SI_Figs 3d, 3e, 3f), as mentioned in the answer to point 6, to better understand temporal changes in fire occurrence through the year and how and when it causes higher impacts on jaguars HRs and to overall Pantanal areas (lines 183-190 in the results and lines 297-302, 309-313, 330-336 in the discussion). As we mentioned in point 13 we included references reenforcing the anthropic origin of fires, although the climate scenario (heat and drought) and the abundance of inflammable material contributed to the spread of fires (see lines 194-249, 297-300, 385-417).

15. Line 284 introduces the first use of the abbreviation UPRB but this isn't defined until Line 328. Remove explanation of abbreviation from Line 340.

We reorganized the use of the acronym introducing it in the first appearance (line numbers 222-223).

16. Lines 351-353 can be moved to Introduction.

We moved this part to the introduction as suggested (lines 53 to 55).

17. Figures were self-containing and readable with exception of Fig. 2. Consider replacing “Number of areas selected as HRs with PAs burned (%) with another color or line choice. Orange is too close to red in printed manuscript.

We changed it to another color (purple).

Reviewer #3

Remarks to the Author:

The authors examined how changes in the frequency and intensity of wildfires in the Pantanal ecoregion have impacted jaguars. Using remote sensing data, jaguar population density estimates and GPS data, the authors show that the most recent fires of 2020 were the most intense and impacting jaguar populations, their home ranges and the protected areas within the jaguar home ranges. The authors achieve a lot in this paper; not only characterising the fire history over a long time period and the impacts of fire for people and biodiversity, but also characterises the impacts of these fires on jaguars. Based on my knowledge, this is the first work of its kind within the Pantanal ecoregion. I believe this paper will be of interest to range of ecologists, particularly those interested in wildfire impacts.

18 There is a lot of additional work and analyses performed by the authors that didn't make it to the main text (e.g., as exploring relationships between fires and precipitation, river water levels, land-use change, and wetland extent), and I wonder whether it would be best to split the paper. One paper that focus on the characterisation of fires over time, and the implications for conservation, and a second paper focusing only on fire impacts on jaguars. Alternatively, a journal with a longer format that allows for the complete discussion of implications might also be a good option. I am not saying the work is not publishable, I think the authors have done some great work here, I just wonder if the current format and structure is the best option.

The main focus of this paper is the analysis of impacts of fires on jaguars. In fact, we believe that studies quantifying the impacts of fire on biodiversity are still an important research gape.

We had gathered additional data and done some extra analysis, mostly to back up the discussion in the jaguar paper, but some material could go beyond that. Although we agree with the reviewer that it could be possibly included in another paper, we are also aware that some other recently published papers had addressed fire history and their causes and consequences in much more detail than we did.

We believe that the comments of the reviewer and from the editor to consider to include some extra material within the present text were the best of both Worlds. We used that to improve the introduction and mostly the discussion and expanded a little our manuscript. But we made an effort to keep it below the limit of 5000 words. For this reason, we also kept some text in the SI.

19 Ln 61 - 62: Here the authors mention that extent of the Pantanal that was impacted by the 2020 fires compared to the previous years. It was not clear if this a result of your work, or based on previous work. In results (Ln 115), the same numbers are mention, so I assume these are results from your study. If this is the case, I am not sure I would mention these values in the introduction to avoid duplication and to avoid starting

We adjusted that in the introduction, removing our results (lines 64-66) and kept just the referenced information.

20 The authors estimated home ranges for 48 resident jaguars monitored between 2005 and 2016, and then compared the extent of the home ranges that the fires burnt for each year. It wasn't clear to me how the authors estimated the impacts of the fires on home ranges from 2017 to 2020, without GPS data from those years. Did they assume the 48 residents maintained their ranges for those years?

We had previously addressed this in brief, but we expanded the explanation in the text for further clarification (lines 95-99, 102-105, 108-112 in the introduction, 182-190 in the results and 571 -576 in the methods, see SI_Figs.S3).

In brief, as we can see in the figures S3a, S3b and S3c in the extended data the number of monitored jaguars in each year vary, and may even be absent in some years. On top of that, the occurrence of fire also varies through the space and time. For that reason, as we showed in these figures, the number of individuals monitored coinciding with fire in the same year is low.

However; the original definition of home ranges (HRs) (Burt 1943, now included), and particularly when it is calculated using the Autocorrelated Kernel Density Estimator - AKDE method (Fleming et al. 2015, also included), stands for a area in which the animal will expend all their lives: "*We therefore define the home range area as a percent coverage region, usually taken to be 95%, of the probability distribution of all possible locations, as determined from the distribution of all possible paths (hereafter, range distribution). This is the same distribution that the conventional KDE approach estimates when its input data are independent. The range distribution addresses the lifetime space requirements of an animal and provides a metric that can be compared across individuals*" (Fleming et al. 2015). Furthermore, the HRs characterize a 2nd order habit selection (Johnson, 1980). Which makes each area selected as a HRs by a jaguar as a potential area for a new HR selection by another jaguar. Based on that, we assumed any area selected as a HR (by individuals classified as residents based on AKDE method), at any point in time, as a HR. Which we compared through the years, even if real time data of HR was absent. Furthermore, the figures plotted with the real time impact of fire in the HRs were quite similar with plots assuming HR as stable areas across the years (see SI_Figs.S3).

21 Fig.1: If the authors have not already done so, I would recommend they check if this figure is colour-blind friendly due to the inclusion of red and green.

We checked the figure in relation to the most extreme colour-blind, and present it below. Although we have got some overlap for red and green, we believe it is still possible to make a distinction between the areas with fire and PAs. But we can change that in case the editors request.

Source: <https://www.color-blindness.com/coblis-color-blindness-simulator/>

Main cited references supporting this letter:

Alvarenga, G. C. et al. Multi-scale path-level analysis of jaguar habitat use in the Pantanal ecosystem. *Biological Conservation* 253, 108900 (2021).

Burt, W. H. Territoriality and Home Range Concepts as Applied to Mammals. *Journal of Mammalogy* 24, 346–352 (1943).

FIRMS, F. I. for R. M. S. FIRMS: Fire Information for Resource Management System. *Google Developers* (2020); <https://developers.google.com/earth-engine/datasets/catalog/FIRMS>

Fleming, C. H. et al. Rigorous home range estimation with movement data: a new autocorrelated kernel density estimator. *Ecology* 96, 1182–1188 (2015).

Jolly, W. M. et al. Climate-induced variations in global wildfire danger from 1979 to 2013. *Nat Commun* 6, 7537 (2015).

Johnson, D. H. The Comparison of Usage and Availability Measurements for Evaluating Resource Preference. *Ecology* 61, 65–71 (1980).

Kumar, S. et al. Changes in land use enhance the sensitivity of tropical ecosystems to fire-climate extremes. *Sci Rep* 12, 964 (2022).

- Libonati, R. et al. Assessing the role of compound drought and heatwave events on unprecedented 2020 wildfires in the Pantanal. *Environ. Res. Lett.* 17, 015005 (2022).
- MapBiomias - Projeto MapBiomias, Mapeamento das áreas queimadas no Brasil (Coleção 1), accessed in 19th April 2021: https://mapbiomas.org/en/colecoes-mapbiomas-1?cama_set_language=en
- Marques, J. F. et al. Fires dynamics in the Pantanal: Impacts of anthropogenic activities and climate change. *Journal of Environmental Management* 299, 113586 (2021).
- Mataveli, G. A. V. *et al.* 2020 Pantanal's widespread fire: short- and long-term implications for biodiversity and conservation. *Biodivers Conserv* 30, 3299–3303 (2021).
- MODIS6 - MODIS Collection 6 NRT Hotspot / Active Fire Detections MCD14DL (2020).
- Morato, R. G. et al. Resource selection in an apex predator and variation in response to local landscape characteristics. *Biological Conservation* 228, 233–240 (2018).

REVIEWERS' COMMENTS:

Reviewer #1 (Remarks to the Author):

The revised manuscript sufficiently addresses the concerns from the first revision and provides stronger context and support for the authors' statements. Thank you for providing a thorough revision.

Dear Dr Grinham, Dr Brooke LaFlamme and Communications Biology editors,

**COMMSBIO-22-3755A, Communications Biology,
De Barros *et al.* "Wildfires disproportionately affected jaguars in the Pantanal "**

We happily thank the editors for the opportunity of publishing our manuscript at the Communications Biology journal! We thank also the reviewers for their previous comments. And we are glad that they become pleased by our efforts to address their suggestions.

We have revised our paper and made additional efforts to comply with the journal format requirements. We tried to follow the editorial requests based on the final revisions instructions, the formatting guidelines and the submission file and editorial policy checklists. We replied point by point in the final revision instructions (related manuscript file). But in summary, the main adjustments were: we re-order the Supplementary Information (SI) references, we made a few adjustments on figures and their headings in the main manuscript, and we also made a few other adjustments following the instructions.

One point we were in doubt was about how to number the extended figures. The main figures are numbered as 1, 2, 3 in the main manuscript. All the other figures appear also in the SI, but we select a few among them to be also part the extended material. For this reason, we left the figures in the extended material with the same legends used in the SI. Please let me know if you prefer that I change these legends to another format.

Another important point to mention is that we have made some alternative suggestions to the brief editor's summary. We explain why that needed to be modified in the comments within the revision instructions file (related manuscript file).

Besides the reviewing process, another point that perhaps is worth to mention is that we wrote to the Springer Nature waivers team as soon as we had submitted the manuscript regarding possibilities of waiver or funding to cover the Article Processing charge. After some consideration they said that we would not need to pay the article-processing charge on this occasion. But that was still conditional to the need of acceptance. Please let me know also, if I should contact them again in the next stage.

Finally, we are very pleased with the opportunity to have our work published in the Communications Biology journal and happy with the improvement resultant from the peer review exercise.

Please, do not hesitate in contact me if you think any adjustment are still required.

Thanks again!

With best wishes,

Alan Eduardo de Barros

REVIEWERS' COMMENTS:

Reviewer #1

Remarks to the Author:

The revised manuscript sufficiently addresses the concerns from the first revision and provides stronger context and support for the authors' statements. Thank you for providing a thorough revision.

We thank the reviewers for all their detailed comments provided, which allowed us to significantly improve the manuscript.